# Consumption of non-sugar sweeteners by Brazilian adolescents and adults in 2017–2018: Socioeconomic distribution and food sources

Luisa Gazola Lage [1☯¤*], Claudia Cristina Gonçalves Pastorello[1‡¤],
Marcos Anderson Lucas da Silva[1‡¤], Vanessa dos Santos Pereira Montera[2‡],
Daniela Silva Canella[2‡], Camila Aparecida Borges[3‡], Maria Laura da Costa Louzada[1☯¤]

1 Nutrition Department, School of Public Health, University of São Paulo, São Paulo, São Paulo, Brazil,
2 Department of Applied Nutrition, Institute of Nutrition, Rio de Janeiro State University, Rio de Janeiro,
Rio de Janeiro, Brazil, 3 Department of Food Science and Technology, Luiz de Queiroz College of
Agriculture, University of São Paulo, Piracicaba, São Paulo, Brazil

☯ These authors contributed equally to this work.
‡ CCGP and MALDS also contributed equally to this work. VM, DSC and CAB are expert consultants.
¤ Current Address: School of Public Health, University of São Paulo, São Paulo, São Paulo, Brazil
* luisagazola@usp.br

## Abstract

Non-sugar sweeteners are food additives frequently used as sugar substitutes. Recent evidence has demonstrated harmful health effects of non-sugar sweeteners. However, studies on the consumption of these substances are scarce. This study aimed to describe the consumption of non-sugar sweeteners by Brazilian adolescents and adults in 2017–2018, using data from the most recent nationally representative dietary survey. This cross-sectional study analyzed individual food intake data from the 2017–2018 Household Budget Survey, collected using a 24-hour dietary recall. We assessed the prevalence of consumers of: (i) tabletop sweeteners, (ii) diet/light-labeled products containing non-sugar sweeteners, (iii) non-sugar sweeteners present in foods, and (iv) any source of non-sugar sweeteners. Prevalence estimates were stratified by sex, age group (adolescents ≥10 to <18 years, adults 18 to <60 years, and elderly ≥60 years), and income quintiles. Additionally, food sources, types of sweeteners, and patterns of concurrent sweetener consumption were evaluated. Overall, 20.6% of Brazilians consumed non-sugar sweeteners. Prevalence was higher prevalence among women (22.7%), older adultes (27.0%), adolescents (24.4%), and individual in the highest income quintile (30.7%). Tabletop sweeteners were the main source (38.2%), followed by artificially sweetened beverages, including juices (25.3%), and yogurts or dairy drinks (25.1%). Most individuals consumed a combination of sweeteners, primarily acesulfame K, sucralose, and aspartame. Non-sugar sweeteners are widely consumed in Brazil, particularly among specific demographic groups, with tabletop sweeteners and sweetened beverages as the

**Data availability statement:** This study used data from the Brazilian Household Budget Survey, collected by the Brazilian Institute of Geography and Statistics, that are publicly available and de-identified. Consent was formally recorded by the Instituto Brasileiro de Geografia e Estatística. https://www.ibge.gov.br/estatisticas/sociais/saude/24786-pesquisa-de-orcamentos-familiares-2.html The minimal anonymized dataset necessary to replicate the study findings has been deposited in the Open Science Framework (OSF) repository and is publicly available at the following link: https://osf.io/4ngq2/overview.

**Funding:** The Brazilian public agency " Foundation Coordination for the Improvement of Higher Education Personnel (CAPES)" financed scholarships for authors pursuing a doctorate (LGL, MALdaS, CCGP). The agency did not interfere in the manuscript development process. https://www.gov.br/capes/pt-br.

**Competing interests:** The authors have declared that no competing interests exist.

main sources. These findings provide essential insights to guide public health policies and regulatory discussions.

## Introduction

Excessive consumption of free sugars is consistently associated with body weight gain and the development of non-communicable diseases (NCDs) [1]. Based on this, in 2015, the World Health Organization (WHO) published the *Guideline: sugars intake for adults and children* [2]. This document defines recommendations for maximum sugar consumption limits and guides the implementation of public policies to reduce excessive intake [2]. Suggested measures include taxing sweetened beverages [3] and introducing front-of-pack nutrition labels that indicate excessive amounts of sugar [4]. The push to reduce sugar consumption contributed to the increase in sales of products containing non-sugar sweeteners, which are sweet-tasting substances with low or no calories such as aspartame, acesulfame K, saccharin and sucralose [5,6]. These products include tabletop sweeteners, food products labeled *"diet"* and *"light"*, and other products with added non-sugar sweeteners. However, a growing body of research has observed negative health outcomes resulting from the consumption of non-sugar sweeteners, even when consumed within established safety standards [5]. Intestinal dysbiosis [7,8], cancer [9,10], metabolic changes and weight gain [11] are examples of the potential risks linked to non-sugar sweeteners.

Corroborating this evidence, in 2023, the WHO published a new guideline that does not recommend the use of non-sugar sweeteners by adults, children and pregnant women as a strategy for weight control or to reduce the risk of developing NCDs [5]. This guideline establishes that these substances do not demonstrate long-term efficacy in reducing the risk of weight gain and may pose potential health risks [5]. Therefore, monitoring the consumption of these substances is essential and should guide future research and the development of public policies to protect consumers.

A study carried out in Brazil in 2017, evaluated labels of products sold in five large supermarket chains and found that 10.8% of them contained non-sugar sweeteners in their ingredient lists [12]. Food labels are important tools for assessing the supply of non-sugar sweeteners. However, there is still a shortage of studies, particularly with national representation, that assess the actual consumption of these substances by the population. Therefore, the objective of this study was to describe the consumption of non-sugar sweeteners by Brazilian adolescents and adults in 2017/2018, using the most recent nationally representative dietary survey.

## Methods

### Data source and sampling

This is a cross-sectional study, which used data from the individual food consumption module of the most recent *Pesquisa de Orçamentos Familiares – POF* (Household Budget Survey), carried out by the Brazilian Institute of Geography and Statistics (IBGE), from July 2017 to July 2018 [13].

The 2017–2018 *POF* has national scope, with sampling carried out using a two-stage cluster sampling plan. The primary sampling units (PSUs) were selected by sampling with probability proportional to the number of households in the sector within each final stratum, composing the master sample. The subsample of PSUs for the 2017–2018 *POF* was selected by simple random sampling in each stratum [14].

The information from the *2017–2018 POF* was obtained directly from the selected permanent private households, through interviews conducted by IBGE Regional Teams with their residents, over a period of nine consecutive days. Electronic equipment (tablets) was used for the interviews and for daily records by the interviewee [14].

The individual food consumption module of the 2017–2018 *POF* was applied to a subsample of households (20.112 out of 57.920), randomly selected from the original sample. The module was applied to each of the residents aged 10 or over, belonging to the existing consumption unit(s) in the selected households, totaling 46,164 individuals. For all participants who completed The Individual Food Consumption Module, socioeconomic data such as sex, age, and per capita income were also collected [13].

## Food consumption data collection

Food consumption was assessed through 24-hour recalls. Individuals were asked, in face-to-face interviews, about all foods and beverages (including water) consumed the previous day. The names of the foods consumed, the type of preparation, the measurement used, the quantity consumed, as well as the time and place of consumption (i.e., whether at home or away from home), were provided in detail. Information of the underlying ingredients of culinary preparations was obtained from the Brazilian Food Composition Table (*Tabela Brasileira de Composição de Alimentos – TBCA,* in portuguese) of the University of São Paulo, Food Research Centre, Version 7.0 (Available at: http://www.fcf.usp.br/tbca).

## Estimation of non-sugar sweeteners in foods

The *2017–2018 POF* food consumption database does not provide information on the list of ingredients for each reported food item. Therefore, information on the presence of non-sugar sweeteners in each food was obtained from the "Table of Food Additives in Foods Consumed in Brazil" [15]*,* which was detailed in the supporting information (S1 File)*.* The table shows, for each food item reported by people studied by 2017–2018 *POF*, the presence or absence of non-sugar sweeteners, the number of non-sugar sweeteners present in each item and the presence of substances with technological sweetening function.

In summary, the *"Table of Food Additives"* was developed based on three databases: a) Food Consumption Database – Household Budget Survey (individual consumption module) (2017−2018); b) Database of food labels sold in Brazil – 2017 (reporting the ingredient list); c) Database of food brands in Brazil, Euromonitor – 2017 (reporting each brand's share of sales). A total of 2.072 food items were recorded, divided into three groups: group 1, natural and minimally processed foods (e.g., banana); group 2, packaged and ready-to-eat foods (e.g., instant noodles); group 3, culinary preparations or foods with multiple items (e.g., hot dogs), whose ingredients were extracted from standardized recipes. The identification of food additives occurred in three phases: Phase 1 – For items in groups 2 and 3 with a brand (e.g., Tang soft drink), the ingredient list was selected directly from the label database. Phase 2 – For unbranded items, the Euromonitor database was used to identify the most consumed brands in Brazil, and then the ingredient lists were obtained from the label database. Phase 3 – For items in the consumption database that were not identified in the label database, other criteria were established based on the three most consumed brands.

By identifying the ingredient lists of foods in groups 2 and 3, it was possible to identify the presence or absence of non-sugar sweeteners in each of the 2.702 items in the food consumption database. Further details on the methodology are available in the S1 File.

Non-sugar sweeteners were identified in the ingredient lists according to the name and/or International Numbering System (INS) code of the compounds [16]. In 2017, the following non-sugar sweeteners were considered permitted for

use: acesulfame potassium (INS 950), aspartame (INS 951), cyclamates (INS 952), erythritol (INS 968), steviol (INS 960), isomalt (INS 953), lactitol (INS 966), maltitols (INS 965), mannitol (INS 421), neotame (INS 961), saccharin (INS 954), sorbitol (INS 420i), sucralose (INS 955), thaumatin (INS 957) and xylitol (INS 967) [16].

### Consumption of tabletop sweeteners and diet and light foods

To identify the consumption of tabletop sweeteners, a specific variable from the 24-hour recall of the *POF* food consumption database was used. This variable identified, for each food and drink consumed by the participant, whether tabletop sweetener was added at the time of consumption (such as tea, coffee, juice and porridge). Food products labeled diet [17,18] and light [18] were identified when the name of the food items in the consumer database contained one of these terms, for example "diet cola soft drink".

### Statistical analysis

Analyses were performed using the first 24-hour recall. Firstly, were described the prevalence (frequencies) of consumers of I) tabletop sweeteners; II) diet/light products (with non-sugar sweeteners); III) non-sugar sweeteners present in processed foods (including diet and light) and; IV) non-sugar sweeteners from any source. This analysis was performed for Brazil as a whole and according to sex (female and male), age group (adolescent ≥10 to <18 years, adult 18 to <60 years and elderly ≥60 years) and income (per capita quintiles).

Then, the prevalence of non-sugar sweetener consumers was assessed according to the frequency of foods with non-sugar sweeteners consumed per day (1 time; 2 times; 3 times or more) and according to the number of non-sugar sweeteners consumed per day (1–3, 4–6, and 7–9).

Thirdly, the consumption of non-sugar sweeteners was assessed according to the type of substance. For this purpose, the prevalence of consumers for each compound classified as a non-sugar sweetener was determined. Next, a non-sugar sweetener consumption matrix was created to illustrate the frequencies of concomitant consumption of different non-sugar sweeteners. A network graph was constructed to identify the most prevalent mixtures (concomitant consumption) of non-sugar sweeteners. In the graph, the nodes represent the prevalence of consumers for each substance, that is, the larger the node, the greater the prevalence of consumers of that substance. The edges identify when two additives are consumed at the same time, that is, the more people have consumed the substances at the edge's end nodes at the same time, the thicker the edge. Non-sugar sweeteners that presented a consumption frequency greater than or equal to fifteen were included in the network graph.

Finally, we checked the food sources that contributed most to the consumption of non-sugar sweeteners. For this purpose, foods were grouped into categories: artificial juices (boxed juices, concentrated juices, powdered juice), yogurts and dairy drinks (yogurts, fermented milks, chocolate drinks); soft drinks (sodas, flavored waters); desserts and sweets (gelatin, chocolates, ice cream and treats); other drinks (teas, energy drinks, soy drinks); sweet baked goods (breads, cakes, sweet pies, cookies). Next, it was assessed how much each of these food groups contributed to the total number of non-sugar sweeteners consumed by the population. Stata 2014 and R Studio v4.3.2 software were used.

This research was exempted from evaluation by the ethics committee, as registered by the Ethics Appreciation Presentation Certificate (CAAE) 74701323.6.0000.5421, from the School of Public Health of the University of São Paulo – FSP/USP.

## Results

It was observed that, in 2017–2018, 20.6% of the Brazilian population consumed non-sugar sweeteners. It was estimated that 8.4% of Brazilians consumed tabletop sweeteners, 13.4% foods with non-sugar sweeteners, and only 1.6% diet/light food products containing non-sugar sweeteners (Table 1).

**Table 1. Prevalence (%) of consumers of tabletop sweeteners and foods with non-sugar sweeteners according to socioeconomic and demographic characteristics, in individuals aged ≥ 10 years. Brazil, 2017-2018.**

| | Tabletop sweetener | | Diet/light-labelled products (containing non-sugar sweeteners) | | Non-sugar sweeteners present in foods (including diet/ light) | | Non-sugar sweeteners from any source | |
|---|---|---|---|---|---|---|---|---|
| | % | 95% CI | % | 95% CI | % | 95% CI | % | 95% CI |
| **Total** | 8.4 | 7.9: 8.9 | 1.6 | 1.4: 1.8 | 13.4 | 12.8: 14.1 | 20.6 | 19.9: 21.4 |
| **Sex** | | | | | | | | |
| female | 10.4 | 9.7: 11.1 | 1.9 | 1.6: 2.2 | 13.9 | 13.1: 14.7 | 22.7 | 21.8: 23.7 |
| male | 6.2 | 5.7: 6.7 | 1.3 | 1.0: 1.6 | 12.9 | 12.1: 14.7 | 18.3 | 17.4: 19.3 |
| **Age** | | | | | | | | |
| adolescents (≥10 to <18) | 1.4 | 0.9: 2.2 | 0.6 | 0.4: 0.8 | 23.2 | 21.3: 25.2 | 24.4 | 22.5: 26.5 |
| adults (18 to <60) | 7.0 | 6.5: 7.5 | 1.5 | 1.3: 1.8 | 12.2 | 11.5: 13.0 | 18.2 | 17.3: 19.0 |
| elderly (≥60) | 19.5 | 18.1: 21.0 | 2.8 | 2.2: 3.5 | 10.1 | 8.9: 11.5 | 27.0 | 25.4: 28.7 |
| **Per capita family income** | | | | | | | | |
| quintile 1 | 4.7 | 4.1: 5.4 | 0.5 | 0.4: 0.7 | 9.6 | 8.5: 10.9 | 13.9 | 12.7: 15.3 |
| quintile 2 | 6.9 | 6.1: 7.9 | 1.0 | 0.7: 1.4 | 11.6 | 10.4: 12.9 | 17.8 | 16.4: 19.3 |
| quintile 3 | 7.7 | 6.8: 8.7 | 1.0 | 0.8: 1.4 | 12.9 | 11.7: 14.2 | 19.8 | 18.4: 21.3 |
| quintile 4 | 9.3 | 8.1: 10.6 | 1.8 | 1.3: 2.7 | 13.9 | 12.5: 15.4 | 21.8 | 20.0: 23.6 |
| Quintile 5 | 13.9 | 12.5: 15.5 | 3.7 | 3.0: 4.6 | 19.6 | 17.8: 21.7 | 30.7 | 28.5: 32.8 |

The groups with the highest proportion of non-sugar sweetener consumers were women (22.7%), elderly people (27.0%) and people with higher per capita income (30.7%). Almost ¼ (24.4%) of adolescents consumed non-sugar sweeteners of these. Of these, 23.2% consumed food products with non-sugar sweeteners, only 1.4% consumed tabletop sweeteners and 0.6% consumed diet/light foods with non-sugar sweeteners. In contrast, 10.1% of the elderly consumed food products with non-sugar sweeteners, 19.5% consumed non-sugar sweeteners from tabletop non-sugar sweeteners and 2.8% consumed diet/light foods with non-sugar sweeteners (Table 1).

According to Table 2, the frequency of foods with non-sugar sweeteners consumed per day was 1 (one) time for 12.7% of individuals, 2 (two) times for 5.6% and 3 (three) times or more for 2.4%. However, the number of non-sugar sweeteners consumed per day varied, with the majority consuming 1–3 (9.4%) and 4–6 (9,2%) per day.

The prevalence of consumers by types of substances, in decreasing order, was acesulfame K (19.8%), sucralose (15,8%), aspartame (6,0%), saccharine (5.5%), cyclamates (5.5%), sorbitol (0.8%) and the others had a prevalence equal to or less than 0.2% (maltitol, mannitol, steviol, isomalt, lactitol, erythritol, thaumatin, xylitol, neotame). At least 15.0% of individuals consumed acesulfame K and sucralose and more than 5.5% consumed aspartame, saccharin, and cyclamate concomitantly (Fig 1 and S2 Table).

The food sources that contributed most to the consumption of non-sugar sweeteners were, in decreasing order, tabletop sweeteners (38.2%), artificial juices (25.3%), yogurts or dairy drinks (25.1%), soft drinks (3.7%), desserts/sweets (2.9%), other beverages (2.9%) and baked goods (2.0%) (Fig 2).

## Discussion

The present study observed, for the first time, that more than 1/5 of the population of Brazilian adolescents and adults consumed non-sugar sweeteners in 2017–2018. The consumption of non-sugar sweeteners was more prevalent among women, the elderly, adolescents, and individuals from wealthier groups. Among the elderly, we observed that the main source of the additive was tabletop sweetener, however, the consumption of non-sugar sweeteners among adolescents was marked by ultra-processed foods.

**Table 2. Frequency (%) of consumption of foods with non-sugar sweeteners and the number of non-sugar sweeteners consumed per day, in individuals aged ≥ 10 years. Brazil, 2017-2018.**

|  | % | 95% CI |
|---|---|---|
| **Number of foods with non-sugar sweeteners were consumed per day** | | |
| 1 time | 12.7 | 12.1: 13.2 |
| 2 times | 5.6 | 5.2: 6.0 |
| 3 times or more | 2.4 | 2.1: 2.6 |
| **Number of non-sugar sweeteners consumed per day** | | |
| 1 a 3 | 9.4 | 8.9: 9.9 |
| 4 a 6 | 9.2 | 8.7: 9.7 |
| 7 a 9 | 1.5 | 1.3: 1.7 |

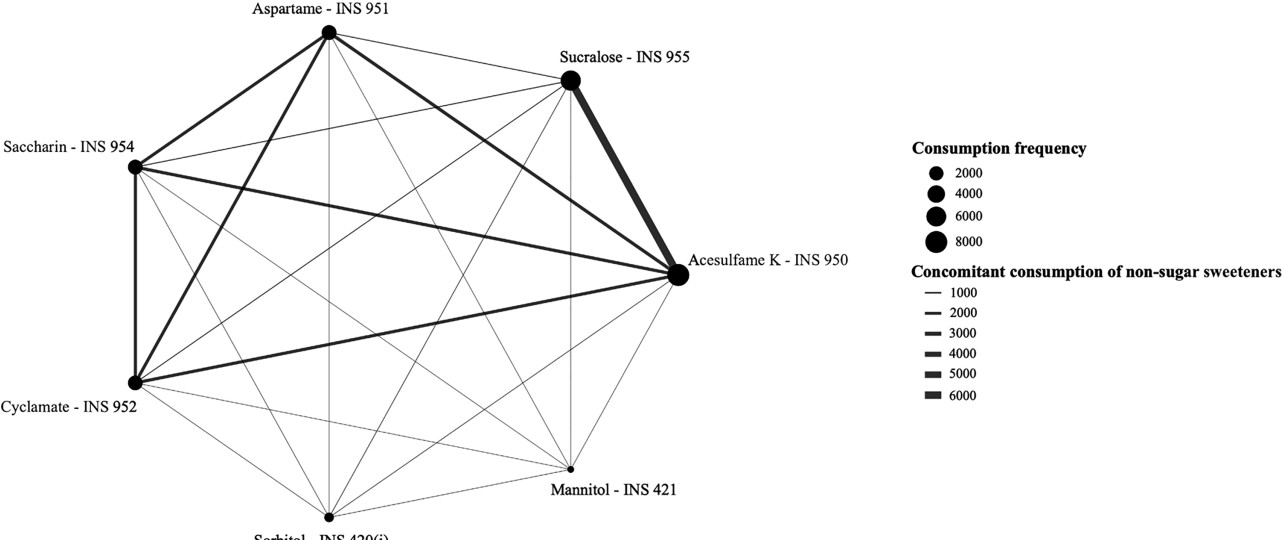

**Fig 1. Network visualization of concomitant consumption between non-sugar sweeteners consumed by individuals aged ≥10 years.** Brazil, 2017-2018.

The contrasting profiles of non-sugar sweetener consumption between adolescents and the elderly observed in this study may reflect different historical and sociocultural contexts. Among the elderly, the higher prevalence of tabletop sweetener use could be attributed to a long-standing history of recommendations emphasizing sugar and calories reduction as part of chronic disease management, rooted in a more traditional "nutritionism" approach [19]. In contrast, the preference for food products with non-sugar sweeteners among adolescents may be driven by modern trends emphasizing aesthetics and "health-conscious" choices, aligning with the growing market of sugar-free products targeted at younger demographics [20]. Also, these choices are likely to be "unconscious", where the individual is unaware of the presence of the sweetener in the food.

Despite the low daily frequency of consumption of foods with non-sugar sweeteners observed in this study (once a day), these foods often contain a mixture of additives with the same technological function, leading to exposure to different substances. In a study that assessed patterns of food additives in food products available in Brazil, sweeteners were identified in two different patterns, one in co-occurrence with flavouring agent, colouring, antihumectant, flavour enhancer,

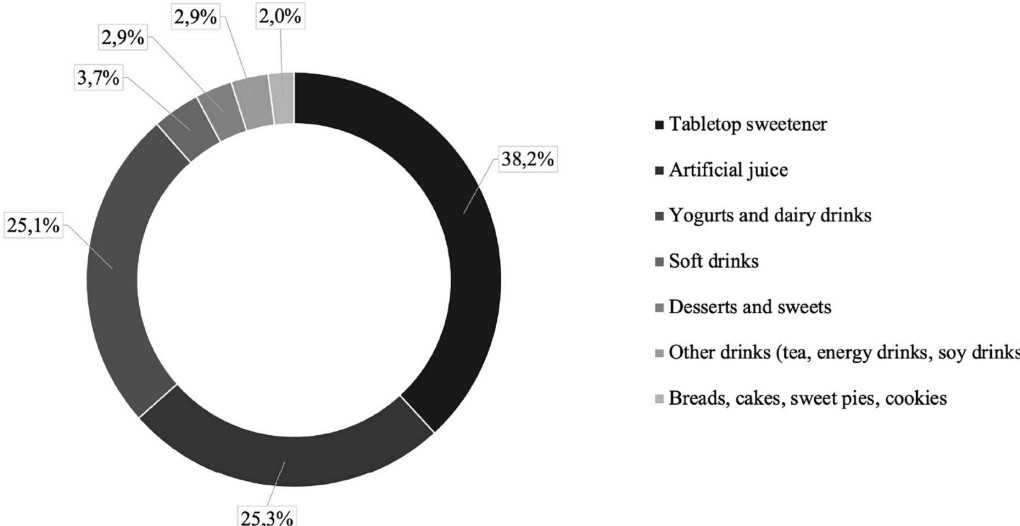

**Fig 2. Relative contribution of each food to total intake of non-sugar sweetener in 2017/2018, in Brazil.**

acidity regulator and maltodextrin-polydextrose (i.e., beverages, candies and desserts) and another one, composed by different sweeteners (i.e., beverages) [12].

To add non-sugar sweeteners to food, the substance must be approved by standards and regulations [21] and present the quantity considered safe for each food or food group, after having undergone toxicity tests and depending on the volume of food consumed by a given population [21]. The *Codex Alimentarius*, published by the Food Agriculture Organization, maintains updated publications that regulate the use of food additives for optional use by member countries of the United Nations [22]. In Brazil, food additives are regulated by the National Health Surveillance Agency, which in turn uses the information published by *Codex Alimentarius*, the European Food Safety Authority (EFSA) and the Food and Drug Administration (FDA) to determine the permitted substances and quantities [23].

However, longitudinal studies have shown negative health outcomes when continuous consumption of non-sugar sweeteners is assessed [7–11]. In the literature, there is still no consensus on the safety of consuming sweeteners. Most studies focus on aspartame, acesulfame K, sucralose, and saccharin due to their prevalence in food and consumption [7]. Research in animals and humans suggests that these four sweeteners can alter gut microbiota and glucose metabolism. However, the methods used in these studies are quite heterogeneous, and some did not show significant results for these outcomes [7]. In a cohort study with an average follow-up of 7.8 years, it was observed that participants who consumed amounts above the median of aspartame, acesulfame K, and sucralose—especially the first two—had a higher risk of developing cancer compared to those who did not consume these sweeteners. On the other hand, a meta-analysis of clinical trials in humans found no association between saccharin consumption and cancer [24].

Children and adolescents are considered even more vulnerable to the negative effects of non-sugar sweeteners. Firstly, because the consumption of these artificial non-sugar sweeteners in childhood can increase preferences for sweet tastes and impact future food choices [25]. Furthermore, the Acceptable Daily Intake (ADI) for non-sugar sweeteners is established in milligrams per kilogram of body weight, meaning that children, due to their lower body weight, may have a relatively higher exposure to these substances compared to adults. This relative overexposure is particularly concerning, as the developing metabolic systems, organs, and tissues of children—depending on their age—may have a reduced capacity to process these compounds effectively, potentially leading to greater vulnerability to adverse effects. Moreover, while the ADI represents a level considered safe for daily consumption over a lifetime, it remains uncertain whether this

safety threshold can be guaranteed when exposure begins at an earlier and more critical stage of life, such as in childhood. Finally, studies that evaluated the consumption of non-sugar sweeteners by children and adolescents remain controversial in the literature, sometimes showing a positive association with a reduction in body measurements, sometimes without significance. The evidence from adult studies (greater in number and scope) suggests that similar effects might also be observed in younger populations [5].

Very few studies with national representation have evaluated the prevalence of non-sugar sweetener consumers. In the USA, 25% of children and 41% of adults consumed non-sugar sweeteners between 2011 and 2012 [26]. In France, a large-scale study conducted between 2017 and 2020 found that three non-sugar sweeteners were among the fifty most prevalent food additives in the study sample, namely sorbitol (18.97%), acesulfame K (12.50%), and sucralose (8.06%) [27]. Finally, in 2011, in the city of Buenos Aires, Argentina, it was observed that 51% of children and adolescents consumed foods with non-sugar sweeteners; aspartame and acesulfame-K were the most prevalent non-sugar sweeteners in the study population [6].

When comparing this study data with the literature, it is possible to observe that each country has specificities regarding the profile of consumers and the non-sugar sweeteners ingested. While in France the most prevalent non-sugar sweeteners were, in decreasing order, sorbitol, acesulfame K and sucralose [27], in Brazil they were acesulfame K, sucralose, aspartame, saccharin, cyclamates and sorbitol. These variations between countries probably occur due to the legislation in force in each location, the food consumption pattern and also economic diferences [28–30]. Furthermore, the timing of data collection is a critical factor. Studies evaluating the use of non-sugar sweeteners in earlier periods, such as 2011, are likely to yield different results compared to studies conducted more recently, like the French study evaluating sweeteners used between 2017 and 2020. Over the years, the types of sweeteners used have shifted, with some becoming more prevalent while others have decreased in use.

It is known that the presence of food additives, mainly "cosmetics" (i.e., those that modify the sensory characteristics of the product such as color, aroma, texture) is an inherent characteristic of ultra-processed foods [31], therefore, a positive relationship is expected between the consumption of ultra-processed foods and non-sugar sweeteners. However, it is hypothesized that countries with the highest average total calories from ultra-processed foods also have the highest prevalence of non-sugar sweetener consumers. While Brazilians in 2017–2018 consumed on average 19.7% of total calories from ultra-processed foods [32], in the USA, in 2015/2016, the percentage rose to 56.2% [33].

Since the consumption of ultra-processed foods has a positive relationship with the income of countries, it is expected that non-sugar sweeteners will also have a positive relationship. Different studies have observed that the group of richer countries consumed a greater volume of UPF and non-sugar sweeteners when compared to middle and low-income countries [28,29]. However, UPF consumption continues to grow in Brazil, more intensely among the lower-income population, which will probably lead to an increase in exposure to non-sugar sweeteners [33].

Understanding the consumption of substances that have potential negative effects on health is essential to develop public policies. In 2022, Resolution No. 429, of October 8, 2020 [34], came into force in Brazil, which established new rules for nutritional labeling, including front-of-pack nutritional labeling that informs the consumer if the food contains excessive amounts of sugar, saturated fat and/or sodium. Since non-sugar sweeteners are important substitutes for sugar, it is expected that many products will be reformulated to avoid reaching the amounts established by law, which requires manufacturers to include the message "excess added sugar" on the label.

That was confirmed in Chile when, after the government implemented the Food Labeling and Advertising Law in 2016 – with a front-of-pack warning on products with high sugar content – there was a 15% increase in new products with non-sugar sweeteners in the list of ingredients [6]. Due to the Chilean experience, in 2020 in Mexico and in 2022 in Argentina, the presence of non-sugar sweeteners was included in the front-of-pack warning package to prevent food reformulations by the industry [35].

This study has limitations arising from potential biases concerning the use of a 24-hour recall, such as underestimation/overestimation of the consumption of certain food groups. To minimize some of these biases, collection instruments were pretested, validated, and subjected to quality control, with inconsistent records excluded and replaced by imputed values. Additionally, information on usual intake is not available, which could be considered a limitation. However, given the large sample size of this national survey, the data provide a robust and representative picture of the population's dietary patterns, mitigating some of the concerns related to individual intake variability.

Furthermore, the presence of non-sugar sweeteners in foods was obtained from a table created from the list of ingredients of products whose brands were the best-selling in Brazil, and not directly from the product consumed. However, we highlight the innovation of the methodology developed in this study that allowed us to estimate, with well-defined criteria, the presence of non-sugar sweeteners in foods. Food composition tables from different countries also assume criteria for the selection of foods with different brands and, despite the existence of biases, when describing the selection process, they are widely accepted by the scientific community.

## Conclusion

This study indicated that a significant portion of the Brazilian population consumed non-sugar sweeteners. Women, the elderly, adolescents and the wealthiest were the most prevalent groups. It was also noted that Brazilians consume a mixture of substances classified as non-sugar sweeteners. Finally, the main dietary sources of this food additive are artificially sweetened beverages or when it is added by the individual via tabletop sweetener.

## Supporting information

**S1 File. Preparation of the Food Additives Table to estimate the non-sugar sweeteners present in foods reported by the 2017–2018 Household Budget Survey.**
(DOCX)

**S2 Table. Total frequency of non-sugar sweeteners consumed in foods and prevalence (%) of consumers of non-sugar sweeteners in foods, according to compounds classified as non-sugar sweeteners, in individuals aged ≥ 10 years.** Brazil, 2017–2018. Note: The non-suger sweeteners Steviol (INS 960), Isomalt (INS 953), Lactitol (INS 966), Erythritol (INS 968), Thaumatin (INS 957), Xylitol (INS), Manitol (421) and Neotame (INS 961) had a very low frequency (≤ 15), so they were not included in the analyses and in the table. [1]International Numbering System for Food Additives (INS).
(DOCX)

## Author contributions

**Conceptualization:** Luisa Gazola Lage, Maria Laura da Costa Louzada.

**Data curation:** Luisa Gazola Lage, Claudia Cristina Gonçalves Pastorello, Marcos Anderson Lucas da Silva, Vanessa dos Santos Pereira Montera, Daniela Silva Canella, Camila Aparecida Borges, Maria Laura da Costa Louzada.

**Formal analysis:** Luisa Gazola Lage, Claudia Cristina Gonçalves Pastorello, Marcos Anderson Lucas da Silva, Maria Laura da Costa Louzada.

**Investigation:** Luisa Gazola Lage, Maria Laura da Costa Louzada.

**Methodology:** Luisa Gazola Lage, Claudia Cristina Gonçalves Pastorello, Marcos Anderson Lucas da Silva, Vanessa dos Santos Pereira Montera, Daniela Silva Canella, Camila Aparecida Borges, Maria Laura da Costa Louzada.

**Project administration:** Luisa Gazola Lage.

**Supervision:** Maria Laura da Costa Louzada.

**Validation:** Luisa Gazola Lage, Claudia Cristina Gonçalves Pastorello, Marcos Anderson Lucas da Silva, Vanessa dos Santos Pereira Montera, Daniela Silva Canella, Camila Aparecida Borges, Maria Laura da Costa Louzada.

**Visualization:** Luisa Gazola Lage, Claudia Cristina Gonçalves Pastorello, Marcos Anderson Lucas da Silva, Vanessa dos Santos Pereira Montera, Daniela Silva Canella, Camila Aparecida Borges, Maria Laura da Costa Louzada.

**Writing – original draft:** Luisa Gazola Lage, Claudia Cristina Gonçalves Pastorello, Marcos Anderson Lucas da Silva, Maria Laura da Costa Louzada.

**Writing – review & editing:** Luisa Gazola Lage, Claudia Cristina Gonçalves Pastorello, Marcos Anderson Lucas da Silva, Vanessa dos Santos Pereira Montera, Daniela Silva Canella, Camila Aparecida Borges, Maria Laura da Costa Louzada.

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
