## [Decision Letter · Decision Letter 0]

4 Jul 2025

Dear Dr. Lage,

Thank you for submitting your manuscript to PLOS ONE. After careful consideration, we feel that it has merit but does not fully meet PLOS ONE’s publication criteria as it currently stands. Therefore, we invite you to submit a revised version of the manuscript that addresses the points raised during the review process.

We look forward to receiving your revised manuscript.

Kind regards,

Eduardo Augusto Fernandes Nilson, DSc

Academic Editor

PLOS ONE

Journal Requirements:

2. Please update your submission to use the PLOS LaTeX template. The template and more information on our requirements for LaTeX submissions can be found at http://journals.plos.org/plosone/s/latex .

6. We notice that your supplementary table are included in the manuscript file. Please remove them and upload them with the file type 'Supporting Information'. Please ensure that each Supporting Information file has a legend listed in the manuscript after the references list.

Additional Editor Comments:

Thank you for your submission. The manuscript is well-writen and addresses and important public nutrition issue. The peer reviewers have assessed the manuscript and provided several comments and suggestions. We recommend a minor revision.

Reviewers' comments:

Reviewer's Responses to Questions

**Comments to the Author**

1. Is the manuscript technically sound, and do the data support the conclusions?

Reviewer #1: Yes

Reviewer #2: Yes

2. Has the statistical analysis been performed appropriately and rigorously?

Reviewer #1: Yes

Reviewer #2: Yes

3. Have the authors made all data underlying the findings in their manuscript fully available?

Reviewer #1: Yes

Reviewer #2: Yes

4. Is the manuscript presented in an intelligible fashion and written in standard English?

Reviewer #1: Yes

Reviewer #2: Yes

Reviewer #1: Well-designed work of extreme relevance to public health.

Methods:

To make it more specific, the authors could include the question used to identify what the person normally uses for sweetening (sugar, sweetener, both or no sweetener). Was this variable used only to determine the prevalence of non-sugar sweetener? Or was the information on the sweetener from the 24-hour recall also used?

Even though there is a supplement to provide details of the methodology used to identify the type of non-sugar sweetener, it would be important to have a sentence in the article's method succinctly highlighting this process and indicating more details in the supplement). Thus, it could be highlighted that, since the bank does not have, in most cases, the brand of the food products, the Euromonitor (2017) and IDEC (2017) references were used to identify the type of non-sugar sweetener present in the foods mentioned in the 24-hour recall (Appendix I).

The National Food Survey collected two 24-hour recalls. Was any analysis performed to verify whether the second day presented the same pattern of non-sugar sweetener consumption as the first day? It would be interesting to justify the choice of working only with the first day of the 24-hour recall.

Some parts of the discussion were left without reference. For example: “Among the elderly, the higher prevalence of tabletop sweetener use could be attributed to a long-standing history of recommendations emphasizing sugar reduction as part of chronic disease management, rooted in a more traditional "nutritionism" approach.”

“Finally, studies that evaluated the consumption of non-sugar sweeteners by children and adolescents remain controversial in the literature” – The authors could cite some of the controversies.

“in Brazil they were acesulfame K (19.8%), sucralose (15.8%), aspartame (6.0%), saccharin (5.5%), cyclamates (5.5%) and sorbitol (0.8%).” – In the discussion, the values already described in the results should not be mentioned again. In the discussion section, it could simply say “in Brazil they were acesulfame K, sucralose aspartame, saccharin, cyclamates and sorbitol.”

“Countries with the highest average total calories from ultra-processed foods also had the highest prevalence of non-sugar sweetener consumers.” – Give examples of these countries and include the reference.

The discussion could include a paragraph talking about some of the health impacts already documented with the most consumed non-sugar sweeteners in Brazil according to this study (acesulfame K, sucralose aspartame, saccharin, cyclamates and sorbitol).

Reviewer #2: 70 Organization (WHO) published the Guideline: sugars intake for adults and children.(2)) Remove “)’”

73 taxing sweetened beverages(3). Space after (3)

74 excessive amounts of sugar.(4)The push. Space after (4).

76 low or no calories such as aspartame, acesulfame K, saccharin and sucralose. (5,6)These products. Space after (5, 6).

81 weight gain(11). Space after gain.

84 for weight control or to reduce the risk of developing NCDs(5). Normally you are putting the reference number before the paragraph. Why in this case did you put it after the end of the sentence?

89 A study carried out in Brazil in 2017, evaluated labels of products sold in five large supermarket chains and found that 10.8% of them contained non-sugar sweeteners in their ingredient lists. Comma after 2017.

100 out by the Brazilian Institute of Geography and Statistics (IBGE), from July 2017 to July 2018(13) . Normally you are putting the reference number before the paragraph. Why in this case did you put it after the end of the sentence? Space after 2018.

105 sampling in each stratum.(14) Normally you are putting the reference number before the paragraph. Why in this case did you put it after the end of the sentence?

115 income were also collected.(13) Normally you are putting the reference number before the paragraph. Why in this case did you put it after the end of the sentence?

124 7.0 (Available at: http://www.fcf.usp.br/tbca).(15) . Put (Available at: http://www.fcf.usp.br/tbca) just in the references. Not in the text.

143 954), sorbitol (INS 420i), sucralose (INS 955), thaumatin (INS 957) and xylitol (INS 967).(17).

You have already inserted (17) on the beginning of the paragraph.

148 tea, coffee, juice, porridge). Food products labeled diet(18,19). Space after diet.

174 concentrated juices, powdered juice), 174 concentrated juices AND powdered juice),

175 The same from 174.

176 The same from 174.

177 The same from 174.

192 According to table 2. Table 2

192 According to table 2, the frequency of foods with non-sugar sweeteners consumed per day was 1 193 (one). Remove 1 and just use one…..

205 (2.0%) (figure 2). Figure 2. Capital letter.

217 Many current foods contain a mixture of sugar and sweeteners. Another possibility is that teenagers do not check the ingredients and consume them without knowing. For example, the most well-known cola-type soft drink uses a mixture of sweetener and sugar and sweetener without stating on the label that it is a light product. I believe that this information can be included and also emphasized that the results obtained may be underestimated, since the POF does not have information on this.

220 products targeted at younger demographics.(20). Normally you are putting the reference number before the paragraph. Why in this case did you put it after the end of the sentence?

227 another one, composed by different sweeteners (i.e. beverages).(12) Normally you are putting the reference number before the paragraph. Why in this case did you put it after the end of the sentence?

230 undergone toxicity tests and depending on the volume of food consumed by a given population 231 (21) The Codex Alimentarius, published by the Food Agriculture Organization, maintains updated. There is a period missing somewhere in the sentence.

236 permitted substances and quantities. (23) Normally you are putting the reference number before the paragraph. Why in this case did you put it after the end of the sentence?

260 acesulfame-K were the most prevalent non-sugar sweeteners in the study population.(27) Normally you are putting the reference number before the paragraph. Why in this case did you put it after the end of the sentence?

274 of ultra-processed foods (31), therefore, a positive relationship is expected between the. The reference (31) refers to the previous sentence or the following sentence? If refers to the previous sentence, it will be necessary to allocate after comma.

301 of these biases. What is biases?

369 No space

Standardize the reference numbers in the text. I suggest placing them after the sentence, but if you prefer to place them before the sentence, avoid leaving the number separated on a line other than where the paragraph begins.

Example: (19) Paragraph .....

Do not leave it like this: (19)

Paragraph.......

Standardize references. Suggestion: Horie M, Ishikawa F, Oishi M, Shindo T, Yasui A, Ito K. 2007. Rapid determination of cyclamate in foods by solid-phase extraction and capillary electrophoresis. J Chromatogr A. 1154(1-2), 423–428.

**Do you want your identity to be public for this peer review?** For information about this choice, including consent withdrawal, please see our Privacy Policy

Reviewer #1: No

Reviewer #2: **Yes: ** Ícaro Gouvêa Nicoluci

---

## [Author Response · Author response to Decision Letter 1]

17 Aug 2025

Dear Editors and Reviewers,

I thank you for your efforts in reviewing the manuscript and contributing to its improvement. I am pleased that this work met PLOS ONE's criteria, with a thorough evaluation. I would like to point out that the editor and reviewers' comments were pertinent. Below, I will address each of them in detail.

Reply to the editor:

1. Please ensure that your manuscript meets PLOS ONE's style requirements and the PLOS LaTeX template.

The manuscript was formatted according to the guidelines provided by PLOS ONE.

I will review the discrepancies in the system.

3. Data availability statement of the submission form

The authors have committed to pre-organizing the research data so that it is available upon request.Some of the data for this study is available on the following websites:

- Household Budget Survey: https://www.ibge.gov.br/estatisticas/sociais/saude/24786-pesquisa-de-orcamentos-familiares-2.html

- Food Additives Table: https://osf.io/m37qj/wiki/home/

4. Please include your full ethics statement in the ‘Methods’ section of your manuscript file.

The exemption statement for evaluation by the ethics committee was added to the "Methods" section.

“This research was exempted from evaluation by the ethics committee, as registered by the Ethics Appreciation Presentation Certificate (CAAE) 74701323.6.0000.5421, from the School of Public Health of the University of São Paulo - FSP/USP.” (coep@fsp.usp.br)

5. We notice that your supplementary table are included in the manuscript file. Please remove them and upload them with the file type 'Supporting Information'.

The supplementary information has been formatted according to the guidelines.

6. Please review your reference list to ensure that it is complete and correct.

The references were formatted according to the journal's guidelines. Two references were included to meet the reviewers' suggestions. They are:

20. Evert AB, Dennison M, Gardner CD, Garvey WT, Lau KHK, MacLeod J, et al. Nutrition Therapy for Adults With Diabetes or Prediabetes: A Consensus Report. Diabetes Care. 2019;42(5):731-54.

24. Liu L, Zhang P, Wang Y, Cui W, Li D. The relationship between the use of artificial sweeteners and cancer: A meta-analysis of case-control studies. Food Sci Nutr. 2021;9(8):4589-97.

7. Additional considerations

Figure 1 underwent some modifications that the authors considered pertinent to facilitate comprehension. These were:

- The names of the non-sugar sweeteners were removed from the caption because the figure already included them.

- The order of the nodes was arranged in descending order.

The reference "15. Lage LG, Marques TS, Montera V, Canella DS, Borges CA, Martins APB, et al. Table of Composition of Food Additives in Foods Consumed in Brazil in 2017-2018 (TAAB). 1st ed. [e-book]. São Paulo: USP School of Public Health; [in press]" is in the process of being published and is currently under review. The official e-book will be published in Portuguese. The file will be sent with the other documents, named "in press_15..pdf"

Upon initial submission, I requested a waiver of the publication fee, but I received no information as to whether it was granted or denied. Therefore, I emphasize that, if the manuscript is accepted, publication will only be possible if the fee is waived in full.

Reviewer #1:

I appreciate your efforts in reviewing this manuscript. Your input was crucial to improving this work.

1- To make it more specific, the authors could include the question used to identify what the person normally uses for sweetening (sugar, sweetener, both or no sweetener). Was this variable used only to determine the prevalence of non-sugar sweetener? Or was the information on the sweetener from the 24-hour recall also used?

Thank you for identifying gaps that may hinder the reader's understanding. We will add the following information regarding these points:

The consumption of the "table sweetener" was identified as follows:

“For each food item reported in 24-hour recall, there was the option to include the “addition” information for: 01 Olive oil; 02 Butter/margarine; 03 Sugar; 04 Non-sugar Sweetener; 05 Honey; 06 Molasses; 07 Mayonnaise (sauce); 08 Ketchup; 09 Mustard (sauce); 10 Soy sauce; 11 Grated cheese; 12 Heavy cream.”

This variable was used to identify the non-sugar sweetener added by the participant. Thus, it was possible to separately evaluate the tabletop sweetener (added by the participant) and the embedded sweetener (added during processing in packaged food products).

For the reader's better understanding, the sentence in topic 2.4 will be rewritten from “This variable identified, for each participant, whether tabletop sweeteners were added to drinks and dishes at the time of consumption (such as tea, coffee, juice, porridge).” to “This variable identified, for each food and drink consumed by the participant, whether tabletop sweetener was added at the time of consumption (such as tea, coffee, juice, porridge).”

Even though there is a supplement to provide details of the methodology used to identify the type of non-sugar sweetener, it would be important to have a sentence in the article's method succinctly highlighting this process and indicating more details in the supplement). Thus, it could be highlighted that, since the bank does not have, in most cases, the brand of the food products, the Euromonitor (2017) and IDEC (2017) references were used to identify the type of non-sugar sweetener present in the foods mentioned in the 24-hour recall (Appendix I).

Thank you for your observation. More details on the development of the table and the identification of sweeteners have been included in topic 2.3.

2- The National Food Survey collected two 24-hour recalls. Was any analysis performed to verify whether the second day presented the same pattern of non-sugar sweetener consumption as the first day? It would be interesting to justify the choice of working only with the first day of the 24-hour recall.

We chose to use the first 24-hour recall because it was answered by all participants in the sample, while in the second, we had missing data and worse quality of the information collected.

3- Some parts of the discussion were left without reference. For example: “Among the elderly, the higher prevalence of tabletop sweetener use could be attributed to a long-standing history of recommendations emphasizing sugar reduction as part of chronic disease management, rooted in a more traditional "nutritionism" approach.”

Thank you for your observation. We include the reference and we added the term "calories" to the text. “Evert AB, Dennison M, Gardner CD, et al. Nutrition Therapy for Adults With Diabetes or Prediabetes: A Consensus Report. Diabetes Care. 2019 May;42(5):731-754. doi: 10.2337/dci19-0014. Epub 2019 Apr 18. PMID: 31000505; PMCID: PMC7011201.”

“Among the elderly, the higher prevalence of tabletop sweetener use could be attributed to a long-standing history of recommendations emphasizing sugar and calories reduction as part of chronic disease management, rooted in a more traditional "nutritionism" approach.”

4- “Finally, studies that evaluated the consumption of non-sugar sweeteners by children and adolescents remain controversial in the literature” – The authors could cite some of the controversies.

We've taken your suggestion into consideration to make the information clearer. We've added the following excerpt:

“Finally, studies that evaluated the consumption of sugar-free sweeteners by children and adolescents remain controversial in the literature, sometimes showing a positive association with a reduction in body measurements, sometimes without significance”

5- “in Brazil they were acesulfame K (19.8%), sucralose (15.8%), aspartame (6.0%), saccharin (5.5%), cyclamates (5.5%) and sorbitol (0.8%).” – In the discussion, the values already described in the results should not be mentioned again. In the discussion section, it could simply say “in Brazil they were acesulfame K, sucralose aspartame, saccharin, cyclamates and sorbitol.”

Thank you for your observation. The values were removed removed from the discussion.

6- “Countries with the highest average total calories from ultra-processed foods also had the highest prevalence of non-sugar sweetener consumers.” – Give examples of these countries and include the reference.

Thank you for this important observation. We chose to rewrite the sentence, since there is not enough data to confirm the correlation.

“Houever, it is hypothesized that countries with the highest average total calories from ultra-processed foods also have a higher prevalence of non-sugar sweetener consumers. While Brazilians, in 2017-2018, consumed an average of 19.7% of their total calories from ultra-processed foods (32), in the US, in 2015-2016, the percentage rose to 56.2%. (33)”

7- The discussion could include a paragraph talking about some of the health impacts already documented with the most consumed non-sugar sweeteners in Brazil according to this study (acesulfame K, sucralose aspartame, saccharin, cyclamates and sorbitol).

Since this information is included in the manuscript's introduction, we felt it would be repetitive to address it in the discussion. Therefore, we wrote a new, more detailed discussion paragraph to help readers understand that there is no consensus on the health effects of sweeteners.

In the literature, there is still no consensus on the safety of consuming sweeteners. Most studies focus on aspartame, acesulfame K, sucralose, and saccharin due to their prevalence in food and consumption [7]. Research in animals and humans suggests that these four sweeteners can alter gut microbiota and glucose metabolism. However, the methods used in these studies are quite heterogeneous, and some did not show significant results for these outcomes [7]. In a cohort study with an average follow-up of 7.8 years, it was observed that participants who consumed amounts above the median of aspartame, acesulfame K, and sucralose—especially the first two—had a higher risk of developing cancer compared to those who did not consume these sweeteners. On the other hand, a meta-analysis of clinical trials in humans found no association between saccharin consumption and cancer [24].

Reviewer #2:

70 Organization (WHO) published the Guideline: sugars intake for adults and children.(2)) Remove “)’”

73 taxing sweetened beverages(3). Space after (3)

74 excessive amounts of sugar.(4)The push. Space after (4).

76 low or no calories such as aspartame, acesulfame K, saccharin and sucralose. (5,6)These products. Space after (5, 6).

81 weight gain(11). Space after gain.

84 for weight control or to reduce the risk of developing NCDs(5). Normally you are putting the reference number before the paragraph. Why in this case did you put it after the end of the sentence?

89 A study carried out in Brazil in 2017, evaluated labels of products sold in five large supermarket chains and found that 10.8% of them contained non-sugar sweeteners in their ingredient lists. Comma after 2017.

100 out by the Brazilian Institute of Geography and Statistics (IBGE), from July 2017 to July 2018(13) . Normally you are putting the reference number before the paragraph. Why in this case did you put it after the end of the sentence? Space after 2018.

105 sampling in each stratum.(14) Normally you are putting the reference number before the paragraph. Why in this case did you put it after the end of the sentence?

115 income were also collected.(13) Normally you are putting the reference number before the paragraph. Why in this case did you put it after the end of the sentence?

124 7.0 (Available at: http://www.fcf.usp.br/tbca).(15) . Put (Available at: http://www.fcf.usp.br/tbca) just in the references. Not in the text.

143 954), sorbitol (INS 420i), sucralose (INS 955), thaumatin (INS 957) and xylitol (INS 967).(17).

You have already inserted (17) on the beginning of the paragraph.

148 tea, coffee, juice, porridge). Food products labeled diet(18,19). Space after diet.

174 concentrated juices, powdered juice), 174 concentrated juices AND powdered juice),

175 The same from 174.

176 The same from 174.

177 The same from 174.

192 According to table 2. Table 2

192 According to table 2, the frequency of foods with non-sugar sweeteners consumed per day was 1 193 (one). Remove 1 and just use one…..

205 (2.0%) (figure 2). Figure 2. Capital letter.

217 Many current foods contain a mixture of sugar and sweeteners. Another possibility is that teenagers do not check the ingredients and consume them without knowing. For example, the most well-known cola-type soft drink uses a mixture of sweetener and sugar and sweetener without stating on the label that it is a light product. I believe that this information can be included and also emphasized that the results obtained may be underestimated, since the POF does not have information on this.

220 products targeted at younger demographics.(20). Normally you are putting the reference number before the paragraph. Why in this case did you put it after the end of the sentence?

227 another one, composed by different sweeteners (i.e. beverages).(12) Normally you are putting the reference number before the paragraph. Why in this case did you put it after the end of the sentence?

230 undergone toxicity tests and depending on the volume of food consumed by a given population 231 (21) The Codex Alimentarius, published by the Food Agriculture Organization, maintains updated. There is a period missing somewhere in the sentence.

236 permitted substances and quantities. (23) Normally you are putting the reference number before the paragraph. Why in this case did you put it after the end of the sentence?

260 acesulfame-K were the most prevalent non-sugar sweeteners in the study population.(27) Normally you are putting the reference number before the paragraph. Why in this case did you put it after the end of the sentence?

274 of ultra-processed foods (31), therefore, a positive relationship is expected between the. The reference (31) refers to the previous sentence or the following sentence? If refers to the previous sentence, it will be necessary to allocate after comma.

301 of these biases. What is biases?

369 No space

Standardize the reference numbers in the text. I suggest placing them after the sentence, but if you prefer to place them before the sentence, avoid leaving the number separated on a line other than where the paragraph begins.

Example: (19) Paragraph .....

Do not leave it like this: (19)

Paragraph.......

Standardize references. Suggestion: Horie M, Ishikawa F, Oishi M, Shindo T, Yasui A, Ito K. 2007. Rapid determination of cyclamate in foods by solid-phase extraction and capillary electrophoresis. J Chromatogr A. 1154(1-2), 423–428.

ANSWERS:

Thanks for the formatting points. I'll correct them and be more careful.

“Biases” is the term used to describe a type of limitation. In the case of the sentence, the term indicates a way to reduce the possibility of instrument error.

We appreciate the suggestions in line 217. Regarding the presence of sugar and non-sugar sweetener in a food, we believe it has been addressed in the paragraph addressing the new front-of-pack nutrition labeling adopted in Brazil, which further encourages the presence of sugar and non-sugar sweetener in a food. We consider the suggestion to address the possibility of "unconscious choice," when the consumer is unaware that the product contains sweeteners in its formulation. We have added the following sentence: “Also, these choices are likely to be "unconscious," where the individual is unaware of the presence of the sweetener in the food.”

Regarding standardized references, the first version can be submitted in free formatting. Now, the formatting recommended by the journal will be used. However, we will pay close attention to the details.

I appreciate your efforts in reviewing this manuscript. Your input was crucial to improving this work.

---

## [Decision Letter · Decision Letter 1]

23 Sep 2025

Consumption of non-sugar sweeteners by Brazilian adolescents and adults in 2017-2018: socioeconomic distribution and food sources.

PONE-D-25-19603R1

Dear Dr. Lage,

We’re pleased to inform you that your manuscript has been judged scientifically suitable for publication and will be formally accepted for publication once it meets all outstanding technical requirements.

Kind regards,

Eduardo Augusto Fernandes Nilson, DSc

Academic Editor

PLOS ONE

Additional Editor Comments (optional):

Thank you for addressing all comments and suggestions from the reviewers.

Reviewers' comments:

Reviewer's Responses to Questions

**Comments to the Author**

Reviewer #2: All comments have been addressed

2. Is the manuscript technically sound, and do the data support the conclusions?

Reviewer #2: Yes

3. Has the statistical analysis been performed appropriately and rigorously?

Reviewer #2: Yes

4. Have the authors made all data underlying the findings in their manuscript fully available?

Reviewer #2: Yes

5. Is the manuscript presented in an intelligible fashion and written in standard English?

Reviewer #2: Yes

Reviewer #2: Line 54- "This cross-sectional study analyzed individual food intake data from the 2017–2018 Household Budget Survey, collected using a 24-hour dietary recall."

Line 61- "Overall, 20.6% of Brazilians consumed non-sugar sweeteners. Prevalence was higher among women (22.7%), older adults (27.0%), adolescents (24.4%), and individuals in the highest income quintile (30.7%)."

Line 89- Cancer and not câncer

Line 127- face-to-face

Standardize "diet/light" or"diet and light"

Table 1- Quintile 2- 6.9 and not 6,9

Line 258 "altodextrin-polydextrose". Correct for "maltodextrin-polydextrose"

Table/figure subtitles are inconsistent:

In "Table 1," all words are capitalized.

In "Table 2" and "Table 3," the captions are mixed.

Figures have captions that are too short, lacking complete explanations, while tables have long titles.

**Do you want your identity to be public for this peer review?** For information about this choice, including consent withdrawal, please see our Privacy Policy

Reviewer #2: **Yes: ** ICARO GOUVEA NICOLUCI

---

## [Editor Report · Acceptance letter]

PONE-D-25-19603R1

PLOS ONE

Dear Dr. Lage,

I'm pleased to inform you that your manuscript has been deemed suitable for publication in PLOS ONE. Congratulations! Your manuscript is now being handed over to our production team.

Kind regards,

on behalf of

Dr. Eduardo Augusto Fernandes Nilson

Academic Editor

PLOS ONE